# On-treatment Serum Mac-2 Binding Protein Glycosylation Isomer (M2BPGi) Level and Risk of Hepatocellular Carcinoma Development in Patients with Chronic Hepatitis B during Nucleot(s)ide Analogue Therapy

**DOI:** 10.3390/ijms21062051

**Published:** 2020-03-17

**Authors:** Ayato Murata, Nozomi Amano, Sho Sato, Hironori Tsuzura, Ko Tomishima, Shunsuke Sato, Kohei Matsumoto, Yuji Shimada, Katsuyori Iijima, Takuya Genda

**Affiliations:** Department of Gastroenterology and Hepatology, Juntendo University Shizuoka Hospital, Shizuoka 410-2295, Japan; perseverance3@hotmail.co.jp (A.M.); n-amano@juntendo.ac.jp (N.A.); sho-sato@juntendo.ac.jp (S.S.); htudura@juntendo.ac.jp (H.T.); tomishim@juntendo.ac.jp (K.T.); syusato@juntendo.ac.jp (S.S.); khmatsu@juntendo.ac.jp (K.M.); yshimada@juntendo.ac.jp (Y.S.); katsu0716@shore.ocn.ne.jp (K.I.)

**Keywords:** chronic hepatitis B, hepatocellular carcinoma, Mac-2-binding protein glycosylation isomer, nucleot(s)ide analog, risk factor

## Abstract

We aimed to analyze the serum level of a novel fibrosis marker, Mac-2-binding protein glycosylation isomer (M2BPGi), and its predictive value for hepatocellular carcinoma (HCC) development in chronic hepatitis B (CHB) under nucleot(s)ide analogue (NA) therapy. Serum M2BPGi levels were quantified in 147 CHB patients at baseline, 48 weeks after starting NA therapy, and at the patients’ last visit. The serum M2BPGi level serially decreased at each time point. During the median follow-up time of 6.6 years, 14 of 147 patients developed HCC. Multivariate Cox proportional hazard analysis demonstrated that high serum M2BPGi at 48 weeks was an independent risk factor for HCC development. A cutoff value of M2BPGi at 48 weeks > 1.5 showed an adjusted hazard ratio = 34.9 (95% confidence interval, 4.3–284.9). The 3- and 5-year cumulative incidence of HCC in patients with low M2BPGi were 0.9% and 4.2%, respectively, whereas those in patients with high M2BPGi were 10.1% and 25.6%, respectively (*p* < 0.001). In conclusion, Serum M2BPGi level at 48 weeks is a useful predictor for HCC development in patients with CHB who receive NA therapy.

## 1. Introduction

Persistent hepatitis B virus (HBV) infection is a major global health problem and an estimated 400 million individuals worldwide are HBV carriers. Among them, approximately 1 million people die annually due to progression into decompensated cirrhosis and/or development of hepatocellular carcinoma (HCC) [1,2]. Currently, oral administration of nucleot(s)ide analogues (NAs) are the most popular treatment strategy for patients with chronic hepatitis B (CHB) because of their excellent virologic efficacy and safety profile. Long-term administration of NAs suppresses HBV replication in most patients, resulting in biochemical remission and histological improvement, including the regression of fibrosis and cirrhosis [3,4]. In addition, NA therapy has been reported to reduce the risk of HCC development in patients with CHB. The long-term effect of lamivudine (LAM), the first generation of NAs, was demonstrated in a prospective placebo-controlled trial in 2004. In this study, the risk of HCC development was reduced by 55% in the LAM group, compared to a placebo group [5]. In another study, entecavir (ETV)-treated patients showed a 63% reduction in HCC risk, in comparison to propensity score-matched historical control patients who did not receive antiviral therapy [6]. Thus, the preventive effects of NAs on HBV-related HCC have been demonstrated. However, the risk of HCC development persists even under NA therapy. Several factors, such as older age, male sex, and hepatic fibrosis, have been reported to predict the risk of HCC development in patients with CHB. Among these factors, the existence of advanced hepatic fibrosis or cirrhosis prior to treatment is widely recognized as a significant risk factor for HCC development in patients with CHB [7].

Liver biopsy has been considered the gold standard for assessing the severity of liver fibrosis and cirrhosis [8], although sampling errors and intra- and interobserver variability can lead to staging errors [9,10]. In addition, it is difficult to perform liver biopsy for all patients due to its invasiveness and rare but potentially life-threatening complications [8]. As a result, several alternative laboratory liver fibrosis indices have been proposed. Recently, a new glycol marker for liver fibrosis was developed using the glycan sugar chain-based immunoassay. Mac-2-binding protein glycation isomer (M2BPGi) was identified as a fibrosis-related glycol-alteration [11], and a significant association between its serum levels and histological hepatic fibrosis was reported in chronic liver diseases [12]. Serum M2BPGi level can be quantified in a small serum volume (10 µL) using an automatic and high throughput assay [11]. This advantage over liver biopsy suggests the clinical usefulness of M2BPGi for assessing the risk of HCC development. The aim of this study was to evaluate factors that affect the occurrence of HCC in patients with CHB during NA therapy, with a special focus on the change and its predictive value of baseline and on-treatment serum M2BPGi levels.

## 2. Results

### 2.1. Baseline Patients’ Characteristics

The enrolled 147 patients (93 male and 54 female) had a median age of 55 years (range, 20 to 82 years). The median serum hepatitis B surface antigen (HBsAg) level was 2800 IU/mL (range, 0.07 to 125,000 IU/mL), and the median serum HBV-DNA level was 6.5 LC/mL (range, <2.1 to >9.0 LC/mL). HBeAg positivity was shown in 75 of 147 patients (51.2%). The median HBcrAg level was 5.6 logU/mL (range, <2.9 to >7.0 logU/mL). The baseline biochemical data were median alanine aminotransferase (ALT), 63 IU/L (range, 14 to 1712 IU/L); median albumin, 4.1 g/dL (range, 2.3 to 4.9 g/dL); median platelet count, 17.8 × 10^4^/mL (range, 2.8 to 38.9 ×10^4^/µL), and median M2BPGi, 1.06 cut-off index (COI) (range, 0.32 to 17.95 COI). Relationships between serum M2BPGi levels and patients’ characteristics at baseline are shown in Figure 1. Serum M2BPGi levels were positively correlated with the patients’ age and serum HBcrAg levels, and were negatively correlated with serum albumin. Serum M2BPGi levels in HBeAg-positive patients were significantly higher than those in HBeAg-negative patients. Of the 147 patients, 32 (21.8%) patients were histologically or clinically diagnosed as liver cirrhosis at baseline. The NA therapy was initiated as follows: 107 (72.8%) ETV monotherapy, 36 (24.5%) LAM monotherapy, 2 (1.4%) tenofovir disoproxil fumarate (TDF) monotherapy, one (0.7%) adefovir dipivoxil (ADV) monotherapy, and 2 (1.4%) combination with LAM and ADV.

### 2.2. Virologic and Biochemical Response during NA Treatment

During the median follow-up of 6.6 years (range, 1.1 to 10.9 years), there were 15 NA switch cases and 15 NA add-on cases because of virologic breakthrough or suboptimal virologic response. The virologic and biochemical response during NA treatment is summarized in Table 1. The median HBsAg titer was 2800, 2150, and 1100 IU/mL at baseline, 48 weeks, and last visit, respectively. There was a significant difference among them (*p* < 0.001). The serum HBV-DNA levels decreased at 4.0 LC/mL or less in 89.0% of patients at 48 weeks and reached below the lower limit of quantification in 86.2% of patients at the time of the last visit. Both HBeAg positivity and HBcr levels also decreased during treatment. The percentage of ALT normalization (<40 IU/L) achieved in 88.1% and 90.6% of patients at 48 weeks and at last visit, respectively. Compared with baseline measurement, serum M2BPGi level was significantly decreased at 48 weeks, and serum M2BPGi level at last visit was significantly lower than at 48 weeks (Figure 2).

### 2.3. HCC Development and Risk Analysis

During the follow-up period, HCC development was observed in 14 (9.5%) of the 147 patients. The estimated cumulative incidences of HCC development were 2.9% and 8.9% at three and five years, respectively. Compared with patients who did not develop HCC, those who developed HCC were older (*p* = 0.001); had a higher rate of cirrhosis at baseline (*p* = 0.007); lower platelet counts at baseline (*p* = 0.010) and 48 weeks (*p* = 0.049); and higher M2BPGi at baseline (*p* = 0.003), 48 weeks (*p* = 0.001), and last visit (*p* < 0.001).

To identify factors associated with HCC development in patients during NA therapy, Cox proportional hazard analysis was performed on baseline patients’ characteristics, and virologic and biochemical data at each time point (Table 2). Univariate analysis revealed that age, serum albumin at baseline and 48 weeks, and serum M2BPGi level at every time point were significantly associated with HCC development. Multivariate analysis was performed for the variables that showed statistical significance or close to significance in univariate analysis, such as age, cirrhosis ststus albumin, platelet count, and M2BPGi at baseline; albumin, platelet count, and M2BPGi at 48 weeks; and HBeAg positivity and M2BPGi at the last visit. Multivariate analysis identified that M2BPGi at 48 weeks was an independent risk factor for HCC development.

### 2.4. Serum M2BPGi Level and HCC Development

Since univariate Cox proportional hazard analysis demonstrated that serum M2BPGi levels at every time point were factors associated with HCC development, we determined the cutoff values of serum M2BPGi levels for predicting the development HCC by receiver operating curve (ROC) analysis (Figure 3). 

The area under the ROC curve of serum M2BPGi at 48 weeks was higher than the values for the baseline and last visit, indicating that serum M2BPGi levels at 48 weeks are most effective for predicting HCC (Table 3). From the ROC analysis, M2BPGi at 48 weeks > 1.5 COI was identified as a cutoff value. Sensitivity, specificity, and negative predictive values of M2BPGi at 48 weeks were higher than those at baseline and last visit. Based on multivariate analysis, cutoff value of M2BPGi at 48 weeks > 1.5 COI showed adjusted HR 34.9 (95% confidence interval 4.3–284.9). Kaplan-Meier plot analysis showed that the three- and five-year cumulative incidence rates of HCC development in patients with M2BPGi at 48 weeks ≤1.5 were 0.9% and 4.2%, respectively, whereas those of patients with M2BPGi > 1.5 COI were 10.1% and 25.6%, respectively (*p* < 0.001, Figure 4).

## 3. Discussion

Chronic infection with HBV is widely recognized as one of the most important risk factors for HCC development. Though several studies demonstrated that potent anti-HBV drugs, and NA administration, reduced the risk of HCC development, HCC still occurred even in patients under NA therapy. In this study, 5-year cumulative incidence of HCC development in CHB patients under NA therapy was 8.9%. Previous reports demonstrated that 5-year cumulative incidence of HCC development in NA-treated patients was reported about 3–4% [5,6]. The reason our study cohort showed higher rate of HCC development than previous studies was not fully clarified; however, the patients’ background, especially age and fibrosis stage, might be different among these studies.

M2BPGi was originally developed as a fibrosis-related glycol-alteration, and a significant relationship between serum M2BPGi level and histological fibrosis stage has been reported in several chronic liver diseases [12]. In addition, a recent report found that M2BPGi was not only a serum surrogate marker of hepatic fibrosis but also a useful predictive marker for HCC development in patients with either chronic hepatitis B or C [13,14,15]. On the other hand, quantifying serum M2BPGi level has low invasiveness, as well as cost- and time-effectiveness. These advantages enable clinicians to measure serum M2BPGi level repeatedly for assessing the risk of HCC development during follow-up. However, changes in serum M2BPGi during antiviral treatment are not fully understood, and the best timing for measuring serum M2BPGi during treatment to obtain the best predictive value is also unclear. 

In this study, it became clear that serum M2BPGi level gradually decreased during NA therapy, which effectively suppressed HBV replication and induced biochemical remission in most of the CHB patients. In addition, we found that serum M2BPGi level at 48 weeks was an independent risk factor for development of HCC in patients with CHB under NA therapy. It was not fully clarified why the M2BPGi level at 48 weeks showed higher predictive performance than at other time points, such as before treatment or at the last visit. Since studies demonstrated that regression of hepatic fibrosis was induced by long-term NA administration [3,4], decline of serum M2BPGi level at the last visit (median follow-up; 6.6 years) was thought to reflect regression of fibrosis. However, NA administration for 48 weeks seems too short to induce fibrosis regression. Therefore, even though there was a significant difference in the value of M2BPGi between baseline and 48 weeks, both were thought to reflect the pre-existing hepatic fibrosis, which has been widely reported as a significant risk factor of HCC development [7,16,17,18]. Recently, factors other than hepatic fibrosis have been found to be associated with serum M2BPGi levels. Especially, ALT, a sensitive indicator of necroinflammatory activity in the liver, showed significant correlation with serum M2BPGi level [14]. Elevation of serum M2BPGi was also reported, even in patients with acute liver injury, which demonstrates evident necroinflammation in hepatic parenchyma but scarce hepatic fibrosis [19]. These observations suggest that serum M2BPGi level is affected by necroinflammatory activity or hepatocyte damage in the liver. The baseline M2BPGi was thought to be under influence of necroinflammation in the liver but 48-week M2BPGi was not, because most patients achieved biochemical remission 48 weeks after NA administration. Therefore, it is possible that M2BPGi at 48 weeks reflects pre-existing hepatic fibrosis more accurately than at baseline. Consistent results were reported in patients with chronic hepatitis C: serum M2BPGi and ALT levels decreased just after hepatitis C virus eradication was achieved and M2BPGi decreased just after viral eradication showed higher predictive values for HCC development than pre-treatment M2BPGi [20].

The main limitation was that this study was retrospectively performed in a single center; therefore, the number of cases of HCC development might be small for the analysis. A future large-scale prospective analysis will be required to validate our results.

## 4. Materials and Methods 

### 4.1. Patients

Between July 2003 and November 2015, 147 CHB patients who started initial NA therapy at Juntendo University Shizuoka Hospital were retrospectively enrolled in this study. NA therapy was indicated according to the guidelines of the Japan Society of Hepatology [21]. Eligible patients were confirmed as being hepatitis B surface antigen (HBsAg)-positive for >6 months. The exclusion criteria were (1) positivity for anti-HCV antibodies; (2) history or serologic evidence of any other chronic liver diseases (i.e., autoimmune hepatitis, primarily biliary cirrhosis, hemochromatosis, and Wilson’s disease); (3) evidence of HCC or any suspicious lesions according to ultrasonography, dynamic computed tomography (CT), or magnetic resonance imaging (MRI) at the time of enrollment; (4) a history of previous treatment for HCC or liver transplantation; and (5) a follow-up period of <1.0 year after initiation of NA therapy. All 147 patients met these inclusion and exclusion criteria. 

The study was approved by the Ethics Committee of Juntendo University Shizuoka Hospital (No. 633, 5 February 2019) and were performed in accordance with the Declaration of Helsinki (as revised in Brazil 2013). Informed consent was obtained in the form of opt-out on the web-site.

### 4.2. Laboratory Investigations

All routine laboratory data were collected immediately before the first NA treatment and serially during the treatment. Serum M2BPGi level was measured using serum samples that had been stored at −20 °C. M2BPGi quantification was performed by an immunoassay using a commercially available kit (HISCL M2BPGi; Sysmex Co., Kobe, Japan) and a fully automatic immunoanalyzer (HISCL-5000; Sysmex Co.). Serum HBV viral load was determined using a COBAS TaqMan HBV test v2.0 (Roche Diagnostics, Branchburg, NJ), which has a dynamic range of 2.1–9.0 log copies (LC)/mL. Quantitative measurements of HBsAg, Hepatitis B e-antigen (HBeAg), and hepatitis B core-related antigen (HBcrAg) were conducted using commercial chemiluminescent enzyme immunoassay kits.

### 4.3. Patient Follow-Up

Each patient was examined for biochemical and HBV virologic markers, blood counts, and tumor markers at 1–3 month intervals during treatment. Imaging analysis for HCC surveillance was performed at least every 6 months. HCC was diagnosed predominantly by imaging studies, including dynamic CT and MRI. When the hepatic nodule did not show typical imaging features, diagnosis was confirmed by fine-needle aspiration biopsy followed by a histological examination. Last visit was defined as the day of the last examination or HCC diagnosis. Patient follow-up ended on September 1, 2017.

### 4.4. Statistical Analyses

All statistical analyses were performed using IBM SPSS Statistics 24 (IBM SPSS, Chicago, IL, USA). The Mann-Whitney U test was used for continuous variables, and the corrected chi-squared test was used for categorical variables. Univariate and multivariate Cox proportional hazard models were used to evaluate factors that were significantly associated with HCC development. The Kaplan-Meier method was used to analyze the cumulative incidence of HCC development, and differences were tested using the log-rank test. The hazard ratio (HR) and 95% confidence interval (CI) were calculated. A *p*-value <0.05 was considered statistically significant.

## 5. Conclusions

Serum M2BPGi level gradually decreased in patients with CHB who received NA therapy but monitoring its level at 48 weeks is essential to evaluate the risk of HCC development.

## Figures and Tables

**Figure 1 ijms-21-02051-f001:**
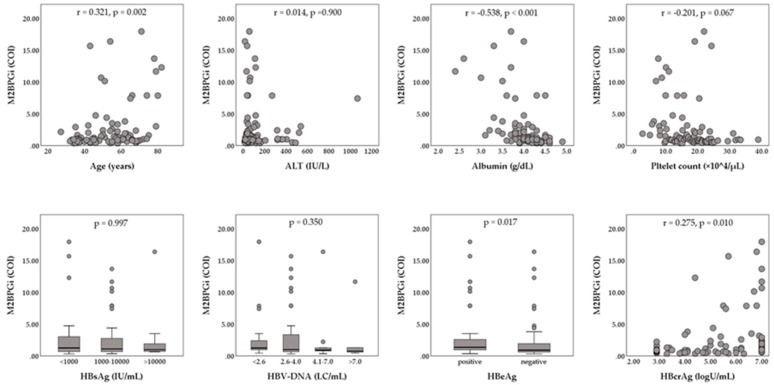
Relationships between Serum levels of Mac-2-binding protein glycosylation isomer (M2BPGi) and patients’ characteristics at baseline. The Mann–Whitney U-test or Kruskal–Wallis test was used for the categorical data. Spearman’s rank correlation coefficient was used for the continuous data. ALT, alanine aminotransferase; COI, cut-off index; HBcrAg, hepatitis B core-related antigen; HBeAg, hepatitis B e-antigen; HBsAg, hepatitis B surface antigen; HBV, hepatitis B virus; LC, log copies.

**Figure 2 ijms-21-02051-f002:**
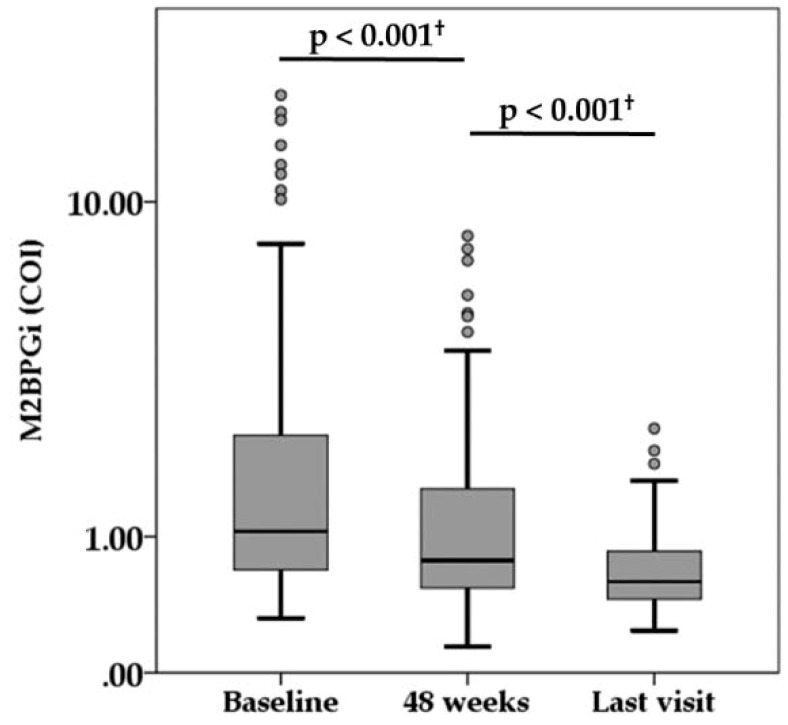
Serum levels of Mac-2-binding protein glycosylation isomer (M2BPGi) during nucleot(s)ide analogue therapy. ^†^ Wilcoxon signed rank test. COI, cut-off index.

**Figure 3 ijms-21-02051-f003:**
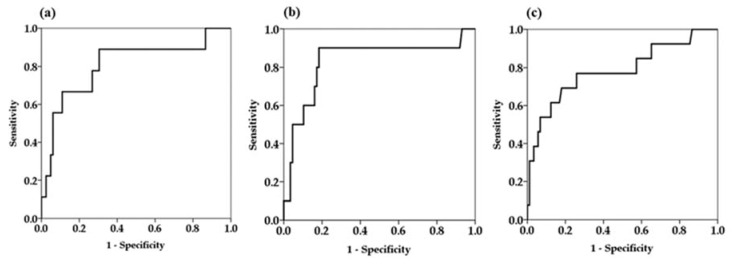
Receiver operator characteristics curves for prediction of hepatocellular carcinoma at different time points. (**a**) Baseline serum Mac-2-binding protein glycosylation isomer (M2BPGi) levels (**b**) serum M2BPGi levels at 48 weeks (**c**) serum M2BPGi levels at the time of last visit.

**Figure 4 ijms-21-02051-f004:**
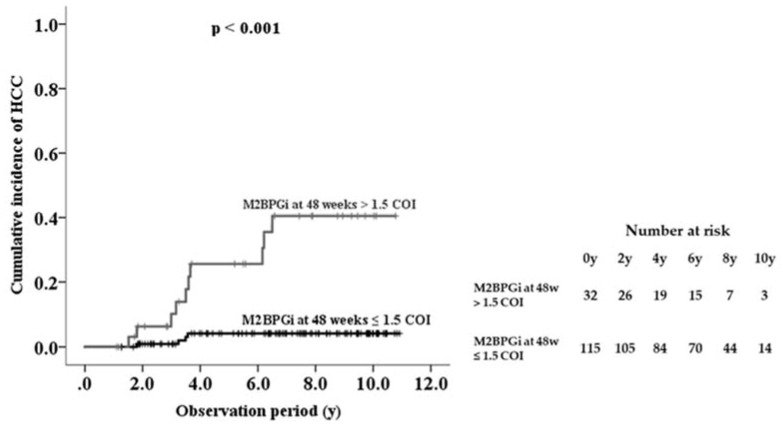
Cumulative incidence of hepatocellular carcinoma (HCC) development in patients with chronic hepatitis B during nucleot(s)ide analogue therapy, shown according to serum Mac-2-binding protein glycosylation isomer (M2BPGi) level at 48 weeks.

**Table 1 ijms-21-02051-t001:** Virologic and biochemical data during NA treatment.

		Baseline	48 Weeks	Last Visit	*p*-Value
HBsAg					
	<1000 IU/mL	23.2%	29.8%	43.1%	<0.001^‡^
	1000–10,000 IU/mL	52.8%	58.8%	48.5%	
	>10000 IU/mL	24.0%	11.4%	8.5%	
HBV-DNA					
	< 2.6 LC/mL	6.8%	75.5%	88.0%	<0.001 ^‡^
	2.6–4.0 LC/mL	10.3%	14.7%	9.8%	
	4.1–7.0 LC/mL	41.8%	9.1%	1.5%	
	>7.0 LC/mL	41.1%	0.7%	0.7%	
HBeAg positivity		51.2%	40.8%	18.3%	<0.001 ^‡^
HBcrAg (logU/mL) ^†^		5.6 (2.9–7.0)	4.6 (2.9–7.0)	3.5 (2.9–7.0)	<0.001 ^§^
ALT (IU/L) ^†^		63 (14–1712)	24 (8–571)	18 (5–173)	<0.001 ^§^
Albumin (g/dL) ^†^		4.1 (2.3–4.9)	4.2 (2.6–4.9)	4.2 (3.0–4.9)	0.004 ^§^
Platelet count (×10^4^/µL) ^†^		17.8 (2.8–38.9)	16.2 (5.7–29.3)	17.2 (5.1–38.2)	0.026 ^§^
M2BPGi (COI) ^†^		1.06 (0.32–17.95)	0.77 (0.14–8.25)	0.59 (0.24–6.57)	<0.001 ^§^

^†^ Data are expressed as medians (range). ^‡^ χ2 test; § Friedman test. *p*-values are for comparisons among data at baseline, at 48 weeks and at last visit. ALT, alanine aminotransferase; COI, cut-off index; HBcrAg, hepatitis B core-related antigen; HBeAg, hepatitis B e-antigen; HBsAg, hepatitis B surface antigen; HBV, hepatitis B virus; LC, log copies; M2BPGi, Mac-2-binding protein glycation isomer; NA, nucleot(s)ide analogue.

**Table 2 ijms-21-02051-t002:** Univariate and multivariate analyses for factors associated with HCC development.

	Univariate		Multivariate	
Variables	HR (95%CI)	*p*-Value	HR (95%CI)	*p*-Value
**Baseline**				
Age	1.08 (1.03–1.13)	0.001		
Male	1.34 (0.42–4.27)	0.622		0.268
Cirrhosis	4.50 (1.58–12.87)	0.005		0.261
HBV-DNA	0.93 (0.72–1.21)	0.585		
HBsAg	1.00 (1.00–1.00)	0.132		
HBeAg positivity	1.68 (0.55–5.13)	0.365		
HBcrAg	1.09 (0.70–1.70)	0.703		
ALT	0.99 (0.98–1.00)	0.178		
Albumin	0.43 (0.16–1.14)	0.089		0.926
Platelet count	0.87 (0.80–0.96)	0.004		0.181
M2BPGi	1.18 (1.07–1.30)	0.001		0.719
**48 weeks**				
HBV-DNA	1.02 (0.73–1.44)	0.903		
HBsAg	1.00 (1.00–1.00)	0.139		
HBeAg positivity	1.03 (0.34–3.14)	0.961		
HBcrAg	0.97 (0.65–1.45)	0.896		
ALT	1.00 (0.99–1.02)	0.937		
Albumin	0.14 (0.04–0.47)	0.002		0.142
Platelet count	0.89 (0.80–0.98)	0.022		0.230
M2BPGi	1.61 (1.28–2.03)	<0.001	1.95 (1.39–2.72)	<0.001
**Last visit**				
NA switch/add-on	0.54 (0.12–2.43)	0.424		
HBV-DNA	1.29 (0.94–1.76)	0.120		
HBsAg	1.00 (1.00–1.00)	0.339		
HBeAg positivity	2.70 (0.84–8.65)	0.094		0.255
HBcrAg	1.22 (0.76–1.96)	0.419		
ALT	1.00 (0.95–1.03)	0.511		
Albumin	0.54 (0.15–1.96)	0.348		
Platelet count	0.93 (0.85–1.02)	0.117		
M2BPGi	1.84 (1.40–2.41)	<0.001		0.858

ALT, alanine aminotransferase; CI, confidence interval; HBcrAg, hepatitis B core-related antigen; HBeAg, hepatitis B e-antigen; HBsAg, hepatitis B surface antigen; HBV, hepatitis B virus; HCC, hepatocellular carcinoma; HR, hazard ratio; M2BPGi, Mac-2-binding protein glycation isomer; NA, nucleot(s)ide analogue.

**Table 3 ijms-21-02051-t003:** Performance of serum M2BPGi levels for predicting HCC.

Time Point	Cutoff	AUROC	Sensitivity	Specificity	PPV	NPV
**Baseline**	1.5 COI	0.805	0.889	0.685	0.889	0.695
**48 weeks**	1.5 COI	0.829	0.900	0.803	0.800	0.804
**Last visit**	0.8 COI	0.816	0.692	0.787	0.692	0.786

AUROC, area under receiver operator characteristic curve; COI, cut-off index; HCC, hepatocellular carcinoma; M2BPGi, Mac-2-binding protein glycosylation isomer; NPV, negative predictive value, PPV, positive predictive value.

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
