# Peer review of "On-treatment Serum Mac-2 Binding Protein Glycosylation Isomer (M2BPGi) Level and Risk of Hepatocellular Carcinoma Development in Patients with Chronic Hepatitis B during Nucleot(s)ide Analogue Therapy"

_ijms, 2020, doi:10.3390/ijms21062051_

Round 1

Reviewer 1 Report

  1. Present HR (CI) of other variables included in the multivariate model in Table 2. 
  2. Liver cirrhosis data is missing in Table 1 and 2. 
  3. I would like to know whether other clinical and laboratory parameters correlate with M2BPGi. Present scatter plots or bar charts of M2BPGi according to continuous and categorical variables, respectively. 

Author Response

Reply to Reviewer #1

Comment 1: Present HR (CI) of other variables included in the multivariate model in Table 2.

Reply to comment 1:

Thank you for your comment. Unfortunately, hazard ratio and 95% confidence interval of statistically non-significant variables are not obtained by stepwise multivariate Cox proportional hazard model. Therefore, we have demonstrated the P-value of all the variables included in the multivariate analysis in the revised manuscript Table 2. We have indicated the variables included in the multivariate analysis as follows (page 4, line122-125): “Multivariate analysis was performed for the variables that showed statistical significance or close to significance in univariate analysis, such as age, cirrhosis ststus albumin, platelet count, and M2BPGi at baseline; albumin, platelet count, and M2BPGi at 48 weeks; and HBeAg positivity and M2BPGi at the last visit.”

Comment 2: Liver cirrhosis data is missing in Table 1 and 2.

Reply to comment 2:

Thank you for bringing this to our notice. According to your suggestion, we have provided information about the baseline cirrhosis status of the study cohort in the “Baseline patients' characteristics” section as follows (page 5, lines 79-80):“Of the 147 patients, 32 (21.8%) were histologically or clinically diagnosed as having liver cirrhosis at baseline.” We have performed statistical analysis including the baseline cirrhosis status and have provided the results in the revised Table 2.

Comment 3: I would like to know whether other clinical and laboratory parameters correlate with M2BPGi. Present scatter plots or bar charts of M2BPGi according to continuous and categorical variables, respectively.

Reply to comment 3:

Thank you for your comment. According to your suggestion, we have demonstrated the relationships between serum M2BPGi levels and baseline patients’ characteristics in the newly added Figure 1.

Reviewer 2 Report

The authors Murata et al. have described reduction  of the fibrosis marker M2BPGi in serum of patients suffering from chronic HBV infection. 

The results are consistent with the view as described with the authors that levels of Mac-2 binding protein can predict development of HCC at a later stage and can be a useful diagnostic tool to predict risk of HCC development.

Figures 2 and 3 are quite hard to read and can be better presented in terms of quality for the reader.

Author Response

Reviewer 2

The authors Murata et al. have described reduction of the fibrosis marker M2BPGi in serum of patients suffering from chronic HBV infection. The results are consistent with the view as described with the authors that levels of Mac-2 binding protein can predict development of HCC at a later stage and can be a useful diagnostic tool to predict risk of HCC development. Figures 2 and 3 are quite hard to read and can be better presented in terms of quality for the reader.

Reply to Reviewer #2:

Thank you for your report. According to your suggestion, we have enhanced image quality in the indicated Figures, and demonstrated them in the revised manuscript as Figure 3 and 4.

Round 2

Reviewer 1 Report

Raised issues have been properly handled and reflected in the revision.